# Effects of Mithramycin on BCL11A Gene Expression and on the Interaction of the BCL11A Transcriptional Complex to γ-Globin Gene Promoter Sequences

**DOI:** 10.3390/genes14101927

**Published:** 2023-10-11

**Authors:** Alessia Finotti, Jessica Gasparello, Cristina Zuccato, Lucia Carmela Cosenza, Enrica Fabbri, Nicoletta Bianchi, Roberto Gambari

**Affiliations:** 1Department of Life Sciences and Biotechnology, Section of Biochemistry and Molecular Biology, Ferrara University, 44121 Ferrara, Italy; jessica.gasparello@unife.it (J.G.); cristina.zuccato@unife.it (C.Z.); luciacarmela.cosenza@unife.it (L.C.C.); enrica.fabbri@unife.it (E.F.); nicoletta.bianchi@unife.it (N.B.); 2Department of Translational Medicine and for Romagna, Ferrara University, 44121 Ferrara, Italy; 3Center “Chiara Gemmo and Elio Zago” for the Research on Thalassemia, Ferrara University, 44121 Ferrara, Italy

**Keywords:** BCL11A, globin genes, β-thalassemia, fetal hemoglobin, mithramycin, transcription, HbF inducers

## Abstract

The anticancer drug mithramycin (MTH), has been proposed for drug repurposing after the finding that it is a potent inducer of fetal hemoglobin (HbF) production in erythroid precursor cells (ErPCs) from β-thalassemia patients. In this respect, previously published studies indicate that MTH is very active in inducing increased expression of γ-globin genes in erythroid cells. This is clinically relevant, as it is firmly established that HbF induction is a valuable approach for the therapy of β-thalassemia and for ameliorating the clinical parameters of sickle-cell disease (SCD). Therefore, the identification of MTH biochemical/molecular targets is of great interest. This study is inspired by recent robust evidence indicating that the expression of γ-globin genes is controlled in adult erythroid cells by different transcriptional repressors, including Oct4, MYB, BCL11A, Sp1, KLF3 and others. Among these, BCL11A is very important. In the present paper we report evidence indicating that alterations of BCL11A gene expression and biological functions occur during MTH-mediated erythroid differentiation. Our study demonstrates that one of the mechanisms of action of MTH is a down-regulation of the transcription of the BCL11A gene, while a second mechanism of action is the inhibition of the molecular interactions between the BCL11A complex and specific sequences of the γ-globin gene promoter.

## 1. Introduction

Induction of increased production of fetal hemoglobin (HbF) is considered a validated approach for therapeutic treatment of β-thalassemia, where the production of adult hemoglobin (HbA) is variably altered by mutations affecting the β-globin gene [1,2,3,4,5]. There is consensus on the fact that HbF production is beneficial for the β-thalassemia patients [4,5]. Consequently, molecules and/or genetic approaches able to stimulate HbF production are of interest in studies aimed at the design and development of novel dev therapeutic protocols for β-thalassemia [6,7,8,9,10,11].

Several studies have demonstrated that an increasing number of well-characterized transcription factors negatively regulate the expression of human γ-globin genes [12,13,14,15,16,17,18,19,20,21,22,23,24,25], such as BCL11A [12,13,14,15], KLF-1 [16,17,18], MYB [19], Oct-1 [20], LYAR [24] and ZNF802 (JAZF1) [25]; all of them are suggested as important direct or indirect repressors of γ-globin gene transcription. Therefore, applied biomedical studies should carefully consider the potential therapeutic effects of targeting these transcription factors to treat hemoglobinopathies, including β-thalassemia and sickle-cell disease [26,27,28,29,30].

The zinc finger transcription factor B-cell lymphoma/leukemia 11A (BCL11A) was shown as the major repressor of HbF expression on the basis of results obtained using several approaches, including genome-wide association studies (GWAS) [31].

The final demonstration of the link between the above mentioned γ-globin gene repressors and the reactivation of HbF is based on gene editing with the objective of disrupting the expression of these transcriptional repressors or their binding site(s) present within the γ-globin genes. In this respect, it has been reported by several studies that genetic disruption of the KLF1, SOX6 and BCL11A genes using the CRISPR/Cas9 system leads to overexpression of the γ-globin genes and HbF reactivation [32,33,34,35,36,37]. For instance, Weber L et al. found that restoration of HbF synthesis can be achieved by CRISPR-Cas9 editing of the γ-globin repressor binding sites present within the promoter of the γ-globin genes [37]. Interestingly, this approach is, at present, under clinical investigation in the NCT03655678 trial, based on the use of CTX001, autologous CRISPR-Cas9 modified CD34^+^ human HSPC (Hematopoietic Stem and Progenitor Cells) [38]. In addition to this and similar genetic approaches, pharmacologically mediated inhibition of the biological activity of transcriptional repressor γ-globin gene expression has been proposed, leading to γ-globin gene activation in vitro, ex vivo and in vivo [6,7]. In this respect, one of the most interesting molecules deserving deep characterization is, in our opinion, the repurposed drug mithramycin (MTH) [39,40,41].

MTH is an antitumor antibiotic approved for the treatment of hypercalcemia [42] and has been also demonstrated to be useful in the treatment of testicular cancer, glioblastoma or Ewing sarcoma [43,44,45]. The firmly established mechanism of action of MTH is based on the binding to GC-rich sequences in DNA, which, in turn, prevents the binding of transcription factors of the SP family to the same gene sequences [46,47]. Mithramycin is, at present, employed in clinical trials on patients with esophageal neoplasms, lung neoplasms, mesothelioma and sarcomas (NCT02859415, NCT01624090, NCT01610570).

Interestingly, mithramycin was found to be a potent inducer of (a) erythroid differentiation in the K562 model system [46] and (b) HbF production in erythroid precursor cells (ErPCs) from normal donors and β-thalassemia patients [48]. The human leukemia K562 cell system was employed as it recapitulates constitutive and induced differentiation and was, in the past, extensively used for the screening of inducers of embryo-fetal globin genes [6,7,49,50]; the ErPCs system was employed as it is a recognized tool for pre-clinical studies of HbF inducers of possible interest for therapeutic protocols for β-thalassemia and other hematological diseases (such as sickle-cell disease) for which an increase in HbF production is beneficial [6,7,48,51].

The main objective of the present study was to verify whether the MTH-mediated induction of differentiation and HbF production are associated with an alteration in the BCL11A gene expression and/or BCL11A biological functions. The effects of MTH on BCL11A gene expression were first analyzed by q-RT-PCR in an ex vivo system constituted by primary erythroid precursor cells (ErPC) from β-thalassemia patients [48]. Second, we analyzed, by Western blotting, the content of BCL11A in MTH treated ErPCs cells, with the aim of determining whether MTH affects BCL11A protein accumulation. Third, we verified, via electrophoretic mobility shift assay (EMSA) [52,53], the possible effect of MTH on the interaction of the BCL11A inhibitory complex with the γ-globin gene promoter.

## 2. Materials and Methods

### 2.1. Human K562 Cell Lines and Culture Conditions

The human leukemia K562 cell line [54] has been purchased from the American Type Culture Collection (ATCC, Rockville, MD, USA); the K562 (BCL11A-XL) clone 12 has been obtained by transfection of K562 cells with the pCDNA3.1-BCL11A-XL vector [55]. The characterization of this clone has been previously reported by Finotti et al. [55], who demonstrated that this clone, despite the fact that it expresses at high level BCL-11A, is, however, inducible by MTH, being, therefore, useful for studies on possible mechanism(s) of action of this DNA-binding drug [55]. The K562 cell line was routinely checked by FACS analysis for expression of erythroid markers, such as Transferrin Receptor (TrfR) and Glycophorin A (GPA) as elsewhere reported [55]. Both K562 cell lines werecultured in humidified atmosphere of 5% CO_2_/air in RPMI 1640 medium (SIGMA, St. Louis, MO, USA) and supplemented with 10% (*v*/*v*) fetal bovine serum (FBS; Biowest, Nuaille, France), 100 U/mL penicillin and 100 µg/mL streptomycin. Mithramycin, cat. M6891, was purchase from Sigma Aldrich (St. Louis, MO, USA). In vitro cell proliferation was studied by determining the cell number per mL with a Z2 Coulter Counter (Beckman Coulter, Fullerton, CA, USA).

### 2.2. Recruitment of β-Thalassemia Patients and In Vitro Culture of Erythroid Precursor Cells (ErPCs)

Cultures of Erythroid Precursor Cells (EpPCs) were derived fromthe peripheral blood of β-thalassemia patients (usually 20 mL of blood) sampled just before transfusion. Mononuclear cells were isolated from peripheral blood samples of β-thalassemia patients by Ficoll-Hypaque density gradient centrifugation and ErPCs isolated as previously described [48,51,56,57]. Patients were recruited at the Day Hospital Thalassemia and Hemoglobinophaties of Azienda Ospedaliera-Universitaria S. Anna (Ferrara, Italy). All the patients participating in this study signed an informed consent form in line with the approval of the Ethical Committee in charge of human studies at the University Hospital. The β-thalassemia patients were all transfusion dependent and were not under therapy with other HbF inducers. For cell culturing, the two-phase liquid culture procedure was employed as previously described [48,57]. Erythroid precursor cell differentiation was assessed by benzidine staining [51].

### 2.3. Preparation of Protein Extracts

The method for preparation of protein extracts has been reported in the study published by Finotti et al. [55]. Briefly, the cellular pellets of PBS-washed, K562 and K562 (BCL11A-XL) cells (2 × 10^5^ cells) were suspended in 50 μL cold water, frozen by dry ice for 5 min and vortexed for 10 s (step repeated 4 times consecutively). At the end of this procedure, the samples were centrifuged at 14,000 rpm for 20 s, and the supernatant cytoplasmic fractions were collected and immediately frozen at −80 °C.

### 2.4. Western Blotting

An amount of 12 μg of cytoplasmic protein extracts were denatured for 5 min at 98 °C in 1× SDS sample buffer and loaded on SDS-PAGE gel (10 cm × 8 cm) in Tris-glycine Buffer. The electrotransfer to 20 microns nitrocellulose membrane (Pierce, Euroclone S.p.A., Pero, Milano, Italy) and the pre-binding treatment of the membranes were performed as described elsewhere [55]. For BCL11A protein detection, the membranes were incubated with BCL11A primary rabbit monoclonal antibody (1:1000) (Cat. A300-382A, Bethyl, Montgomery, TX, USA) and MoAb binding detection, as published elsewhere [55].

After a stripping procedure using the Restore™ Western Blot Stripping Buffer (Pierce), membranes were re-probed with primary antibody against p70S6K, cat. 2708, and Cell Signaling for normalization. X-ray films for chemiluminescent blots were analyzed by Gel Doc 2000 (Bio-Rad Laboratories, Milano, Italy) using a Quantity One program to elaborate the intensity data of our specific target protein. The full procedure can be found in Finotti et al. [55].

### 2.5. Preparation of Nuclear Extracts and Electrophoretic Mobility Shift Assays (EMSA)

Nuclear extracts were prepared as described by Andrews and Faller [58]. Briefly, nuclear proteins were extracted by incubation of purified nuclei for 20 min on ice in 20 mM Hepes/KOH, pH 7.9, 25% glycerol, 420 mM NaCl, 1.5 mM MgCl_2_, 0.2 mM EDTA, 0.5 mM dithiothreitol, 0.2 mM phenylmethanesulfonyl fluoride, 1 μg·mL^−1^ aprotinin, 1 μg·mL^−1^ leupeptin, 2 mM Na_3_VO_4_, and 10 mM NaF (Sigma, St. Louis, MO, USA).

The double-stranded oligonucleotides (ODN) used in the EMSA experiment were designed to mimic the 5′-GGG GCC GGC CGG-3′ (sense strand) sequence present in the γ-globin gene promoter. This sequence has previously been demonstrated to interact with MTH [46] and with the BCL11A transcription factor complex [55]. A total of 3 pmol of ODN were ^32^P-labeled as described [52,53], annealed to an excess of complementary ODN and purified from [γ-^32^P-ATP (Perkin Elmer, Wellesley, MA, USA).

Binding reactions were performed by incubating 2 µg of nuclear extract and 16 fmol of ^32^P-labeled double-stranded ODN and analyzed as elsewhere described [47]. To verify the effects of MTH, DNA probes, or nuclear extracts, were preincubated for 1 h at 4 °C with different concentrations of the compound before the incubation with nuclear extracts. In some experiments, nuclear extracts were incubated with the ^32^P-labeled double-stranded ODN before addition of MTH.

### 2.6. Reverse Transcription and Quantitative Real-Time PCR (RT-qPCR)

For gene expression analysis, 300 ng of total RNA were reverse transcribed using random hexamers. RT-qPCR assay was carried out using gene-specific double fluorescently labeled probes. The nucleotide sequences used for real-time qPCR analysis were 5′-TGG CAA GAA GGT GCT GAC TTC-3′ (γ-globin forward primer), 5′-TCA CTC AGC TGG GCA AAG G-3′ (γ-globin reverse primer) and 5′-FAM-TGG GAG ATG CCA TAA AGC ACC TGG-TAMRA-3′ (γ-globin probe). The kit for quantitative qRT-PCR for BCL11A (Hs00256254_m1) mRNAs was from Applied Biosystems (Applied Biosystems, Monza, Italy). Experimental details of RT-qPCR have been published elsewhere [55].

### 2.7. Statistical Analysis

All the data were normally distributed and presented as mean ± S.D. Statistical differences between groups were compared using one-way ANOVA (Analyses of Variance between groups). The GraphPad Prism 8.2.1 (GraphPad Software Inc., La Jolla, CA, USA) software was employed. Statistical differences were considered significant when *p* < 0.05 (*) and highly significant when *p* < 0.01 (**) and *p* < 0.001 (***). For analysis of correlations between variable, the Spearman test has been employed (https://www.socscistatistics.com/tests/spearman/default2.aspx, accessed on 3 July 2023). rs values > 0.7 and 0.4–0.69 were interpreted as very strong and strong relationships, respectively.

## 3. Results

### 3.1. Mithramycin-Mediated Induction of γ-Globin Genes in Erythroid Precursor Cells (ErPC) Is Associated with a Dramatic Decrease in BCL11A mRNA and Protein Content

When erythroid precursor cells from β-thalassemic patients are cultured in the presence of MTH, we observed, as previously reported [48], an increase in γ-globin gene expression (Figure 1A). In parallel, we observed a strong reduction in the accumulation of BCL11A mRNA (Figure 1B). In these experiments, γ-globin mRNA (Figure 1A) and BCL11A mRNA (Figure 1B) were quantified by RT-qPCR in MTH-treated ErPCs from 11 β-thalassemia patients. This observation was highly reproducible and accompanied by a strong reduction of BCL11A protein production (Figure 1C,D).

These data support the hypothesis that BCL11A network should be considered as one of the major targets of mithramycin activity. This is further sustained by the experiments shown in Figure 2. In panel A of Figure 2, we compared the levels of BCL11A mRNA with the % of increase in HbF analyzed in MTH treated ErPCs from 12 β-thalassemia patients. We found that high relative levels of BCL11A mRNA were associated with low expression of γ-globin genes; conversely, low relative levels of BCL11A mRNA were associated with high induction of HbF. Spearman’s test was conducted to determine the level of correlation between the two variables (BCL11A mRNA and HbF production) and the results obtained suggest that the correlation between the level of BCL11A and the ability of MTH to induce HbF accumulation is very high, being rs = −0.86667 and *p* (2-tailed) = 0.0025 (n = 9) (Figure 2A).

This analysis is in strong agreement with the general conclusion that BCL11A expression inhibits γ-globin gene expression (Figure 2A). In the experiments reported in Figure 2B,C, ErPCs were isolated from two β-thalassemia patients with β^0^39/β^0^39 and β^+^IVSI-6/β^+^IVSI-6 genotypes and induced with increasing concentrations of MTH; after 5 days RNA was isolated and RT-qPCR analysis was performed to detect BCL11A and γ-globin mRNAs. The results obtained show that the increase in γ-globin gene expression is accompanied by a decrease in BCL11A mRNA (Figure 2B,C).

### 3.2. Analysis of the BCL11A Gene Promoter: Presence of Sp1 Transcription Factors Binding Sites Possibly Recognized by MTH

MTH is a DNA-binding drug that is highly selective for G+C-rich region, and, for this reason, is a well-established, strong inhibitor of the interactions between Specific Factor 1 (Sp1) and its binding site(s) present within the promoters of Sp1-regulated genes. Footprinting and EMSA analyses have been employed to show that MTH and Sp1 competitively bind to GC-rich regions present in MTH-regulated gene promoters [59,60]. Accordingly, the ability of MTH to bind to GC-rich DNA and displace Sp1 family proteins from their binding sites has been supported by more than 100 publications [59,60]. For instance, MTH inhibits the binding of SP1 to the promoters of the raptor [61], PAC1 [62], c-myc [63], survivin [64], XIAP [65] and VEGF [66] genes.

Considering this widely recognized mechanism of action, we analyzed the promoter of the BCL11A gene by looking at possible G+C-rich elements. This analysis is reported in Figure 3 and demonstrates the presence of several Sp1 binding sites (red arrows) and two canonical KLF-1 binding sites (blue arrows). It should be considered that Sp1 and KLF-1 are recognized up-regulators of BCL11A gene transcription. Therefore, a reasonable hypothesis is that MTH inhibits the interactions between Sp1 and the BCL11A promoter, thereby inhibiting BCL11A transcription.

### 3.3. A Region of the Human γ-Globin Gene Promoter Recognized by Mithramycin Corresponds to a BCL11A Binding Site: A Second Level of MTH Action?

We have published elsewhere that a region of the human γ-globin promoter bound by MTH corresponds to the BCL11A site present within the γ-globin gene promoter, as published by Bianchi et al. [46] and depicted in Figure 4A. Therefore, we speculated that MTH might also exert its biological activity by interfering with the interactions between the BCL11A complex and the γ-globin mRNAs gene promoter.

Figure 4B shows the effects of an anti-BCL11A antibody on the complexes generated by the interaction between K562 nuclear extracts and the target ^32^P-labelled double-stranded oligonucleotides mimicking the BCL11A γ-globin promoter binding site (underlined in Figure 4A). The data obtained indicate that the addition of the anti-BCL11A antibody suppresses the generation of a protein/DNA complex, indicating BCL11A as a key player in these DNA/protein complexes.

Figure 5 shows an experiment aimed at determining whether MTH is able to interfere with the molecular interactions between K562 nuclear extracts and ^32^P-labelled double stranded oligonucleotides mimicking the BCL11A binding site present within the γ-globin gene promoter.

The results obtained demonstrate dose-dependent inhibition by MTH of the binding of nuclear factors to the ^32^P-labelled double stranded oligonucleotides mimicking the BCL11A binding site. The inhibitory effect was also found when MTH was added after allowing the formation of the protein/DNA complexes, strongly suggesting that MTH is able to disrupt pre-formed interactions.

## 4. Discussion

Mithramycin (MTH) is a DNA-binding drug of increasing interest in therapy. Unlike the analogue chromomycin, MTH reversibly binds target DNA and MTH/DNA complexes are unstable. For this reason, MTH has been demonstrated to have less genotoxicity when compared to chromomycin [46]. MTH has been proposed as an anti-cancer agent and recent published data reinforced this application in oncology, demonstrating that the interest in this compound is still high. MTH is, at present, employed in clinical trials (NCT02859415, NCT01624090, NCT01610570), having as secondary outcomes the evaluation of safety of the treatment. In view of its clinical application and potential interest for other pathologies, MTH was investigated for drug repurposing by several different laboratories, including our research group. In this respect, we have first demonstrated that mithramycin is a powerful inducer of K562 erythroid differentiation [46]; second, we determined its effects on primary erythroid precursor cells isolated from normal donors and β-thalassemia patients, concluding that MTH is one of the most powerful inducers of γ-globin gene expression and HbF production [48]. As firmly established, HbF induction can be considered a very interesting approach for the cure of β-thalassemia and for ameliorating the clinical parameters of sickle-cell anemia [4,5,6,7]. However, it should be considered that MTH (like almost all of the low molecular weight bioactive molecules) has multiple targets in treated cells and that the key targets might be different in different cellular systems and in vivo in different tissues and cellular compartment. For these reasons, studies on the mechanism(s) of action of MTH are relevant.

In the present paper we report evidence supporting the concept that alteration of BCL11A gene expression occurs during MTH-mediated induction of erythroid differentiation of K562 cells and during MTH-mediated upregulation of γ-globin genes in primary Erythroid Precursor Cells (ErPCs). One of the mechanisms of action of MTH is a repression of the transcription of the BCL11A gene (Figure 1, Figure 2 and Figure 3); the second is the inhibition of the interaction between the BCL11A complex and the γ-globin gene promoter (Figure 4). These double effects on the BCL11A network are summarized in Figure 6.

In untreated cells (upper part of Figure 6) the BCL11A gene is highly transcribed and the BCL11A-complex efficiently binds to the γ-globin gene promoter, leading to low transcription of the γ-globin gene and low production of HbF. In MTH-treated cells (lower part of Figure 6) the BCL11A gene is transcribed with low efficiency, due to the MTH-mediated inhibition of the interactions of Sp1/KLF1 to the BCL11A gene promoter (3) and of the BCL11A-complex interaction with the γ-globin gene promoter (see the experiment depicted in Figure 4), leading to lower BCL11A-mediated repression, high transcription of the γ-globin gene and high production of HbF.

One of the limits of our study is the use of only one in vitro cellular model system (the erythroleukemic K562 cell line). This was conducted because this system has been previously employed for determining the effects of MTH [46] and of several other HbF inducers [6,7,49,50,51,67,68]. It will be of interest will be in future studies to employ experimental systems and cell lines derived from patients, such as the HUDEP-1 and HUDEP-2 cell lines, that have been recently used in experimental projects similar to our study [69,70]. Another limit is the number of recruited β-thalassemia patients. In this respect we would like to underline that, in order to limit confounding parameters, we decided to recruit patients not involved in therapeutic protocols with other HbF inducers (such as hydroxyurea and sirolimus) or clinical trials. This strategy limited the number of β-thalassemia patients available for the study. This study should, therefore, be considered a proof-of-principle regarding possible mechanism(s) of action of a drug of possible interest for β-thalassemia.

Concerning possible use of MTH for the therapy of β-thalassemia, one of the limits in the clinical applications of this drug is the severe systemic toxicities, that, among others, include dose-related bleeding and liver toxicity [71]. These unwanted side effects have limited the clinical use of MTH. This has also an impact on proposing MTH in the experimental therapy of hematological diseases needing increased production of HbF.

In order to overcome the limitations of MTH usage in clinical settings, two major different strategies have been proposed so far. The first strategy consists of the generation of structurally related analogues exhibiting improved characteristics affecting activity and safety in comparison to MTH [72,73,74]. The second strategy is based on the use of nanocarriers to reduce the delivery of the compound to non-pathological areas, therefore increasing the therapeutic index [75]. In fact, encapsulation of MTH could improve its pharmacokinetic and biodistribution profile and decrease toxicity without major alterations of biological functions [76,77]. In this respect, Capretto et al. proposed that polymeric micelles delivering MTH and produced by a microfluidic approach warrants further evaluation as a potential therapeutic protocol for β-thalassemia [77].

## 5. Conclusions and Future Perspectives

In this paper, the mechanism of action of the repurposed drug mithramycin (MTH) has been investigated. As summarized in Figure 6, MTH exhibits two activities based on the DNA-binding ability elsewhere described [39,42,44]: (a) inhibition of BCL11A transcription through interference with the BCL11A gene up-regulator SP1 (Figure 3) and (b) inhibition of the interactions of BCL11A to the γ-globin gene promoter (Figure 5). The development of more effective and less toxic MTH analogues and appropriate delivery systems to target selected tissues will be useful fields of investigation to verify possible application of this repurposed drug to β-thalassemia.

The hypothesis that is generated by the present paper is that Sp1 might have a general impact in activating γ-globin genes, since it has been shown that Sp1 binding sites are present in the promoter of several γ-globin gene repressor, such as BCL11A (this paper), KLF1 [78], MYB [79], BACH1 [80], ZBTB7A [81]. Future experiments will clarify whether Sp1 is involved in the regulation of the recently described γ-globin gene repressor LYAR [21,22]. Furthermore, an increasing number of microRNAs targeting Sp1 have been recently described (for instance miR-204, miR-135a-5p, miR-128-3p, miR-335-5p) [82,83,84,85], that should be assayed for possible induction of γ-globin genes through a decrease in the transcription of γ-globin gene repressor caused by miRNA-mediated Sp1 down-regulation.

## Figures and Tables

**Figure 1 genes-14-01927-f001:**
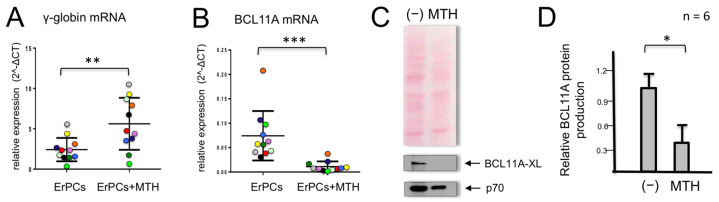
Effects of mithramycin (MTH) on γ-globin and BCL11A mRNAs in erythroid precursor cells (ErPCs) from β-thalassemia patients. ErPCs isolated from 11 β-thalassemia patients (each represented by a different color) were treated for 5 days with EPO in the presence of 20 nM MTH, RNA extracted and RT-PCR for γ-globin mRNA (**A**) and BCL11A-XL (**B**) was performed. (**C**,**D**) ErPCs isolated from six β-thalassemia patients were treated for 5 days with EPO in the presence of 20 nM MTH, proteins were purified and Western blotting was performed using monoclonal antibodies against BCL11A-XL and p70S6K, which were used as the normalization control. An amount of 12 μg of cytoplasmic protein extracts were loaded on SDS-PAGE gel. Representative Western blotting analyses are shown in (**C**), while panel (**D**) summarizes the data obtained in six independent induction experiments. The relative BCL11A protein production values were obtained by densitometry of the autoradiographs. (*): *p* < 0.05 (significant); (**): *p* < 0.01 and (***): *p* < 0.001 (highly significant).

**Figure 2 genes-14-01927-f002:**
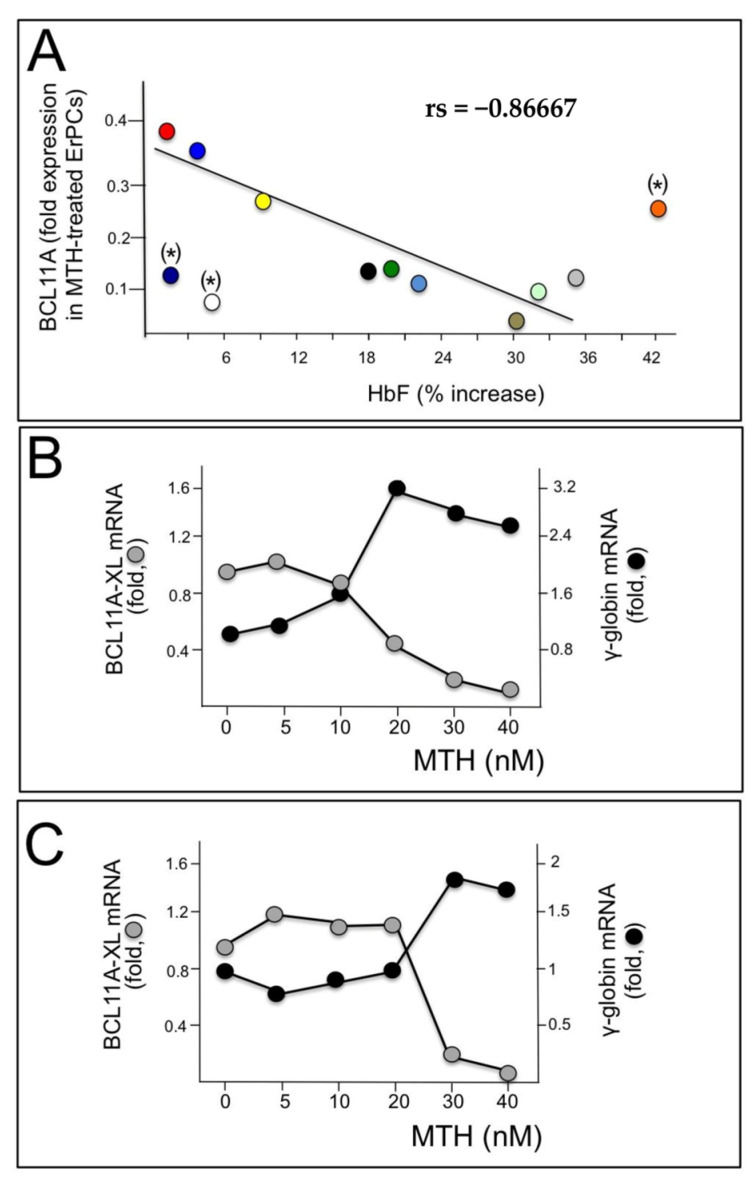
Role of BCL11A in γ-globin gene expression. (**A**) Relationship between % of MTH-induced increase in HbF and BCL11A down-regulation analyzed in ErPCs from 12 patients, each represented by a different color, from which both RNA and proteins were isolated. A good correlation was found in the majority of samples, with the exception of (*)-labelled samples. (**B**,**C**) Relationship between BCL11A and γ-globin gene expression following treatment with different MTH concentrations of ErPCs isolated from two β-thalassemia patients (one β^0^, panel (**B**) and one β^+^, panel (**C**)).

**Figure 3 genes-14-01927-f003:**
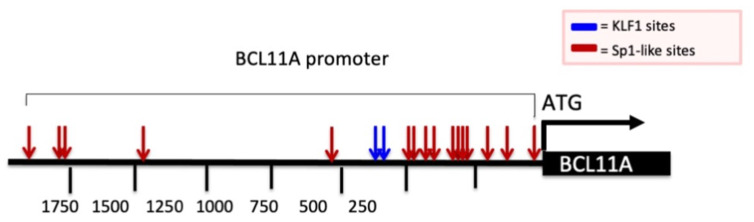
Schematic representation of BCL11A gene promoter and presence of Sp1 and KLF1 binding sites. Red and blue arrows indicate the Sp1 and KLF1 binding sites present in the promoter region of the BCL11A gene. Nucleotide positions are indicated.

**Figure 4 genes-14-01927-f004:**
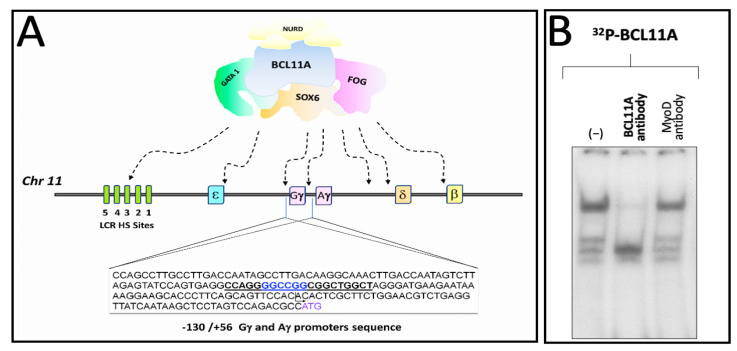
Pictorial scheme of the γ-globin gene promoter and binding of K562 nuclear proteins to the BCL11A consensus sites. (**A**) Underlined sequences 5′-CCAGGGGCCGGCGGCTGGCT-3′ indicate the location of the BCL11A binding sites present in the promoter region of the γ-globin genes. Nucleotide positions are indicated. (**B**) Representative EMSA analysis demonstrating that the BCL11A antibody strongly reduces the interactions between nuclear extracts from the K562-BCL11A (clone #12) [55] and the ^32^P-labelled BCL11A probe.

**Figure 5 genes-14-01927-f005:**
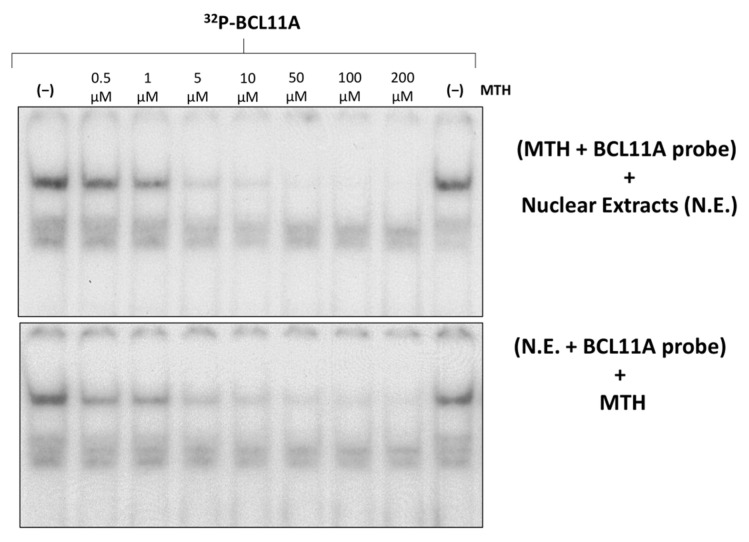
Effects of mithramycin on the binding efficiency of K562 nuclear proteins to the BCL11A consensus sites of the γ-globin gene promoter (BCL11A probe). Effects of increasing concentrations of MTH on binding of K562 nuclear extracts to the ^32^P-labelled BCL11A probe. On the upper site of the panel, the probe was incubated with MTH before the addition of nuclear extracts, while in the experiment on the lower site, nuclear extracts were incubated with the probe before the addition of MTH.

**Figure 6 genes-14-01927-f006:**
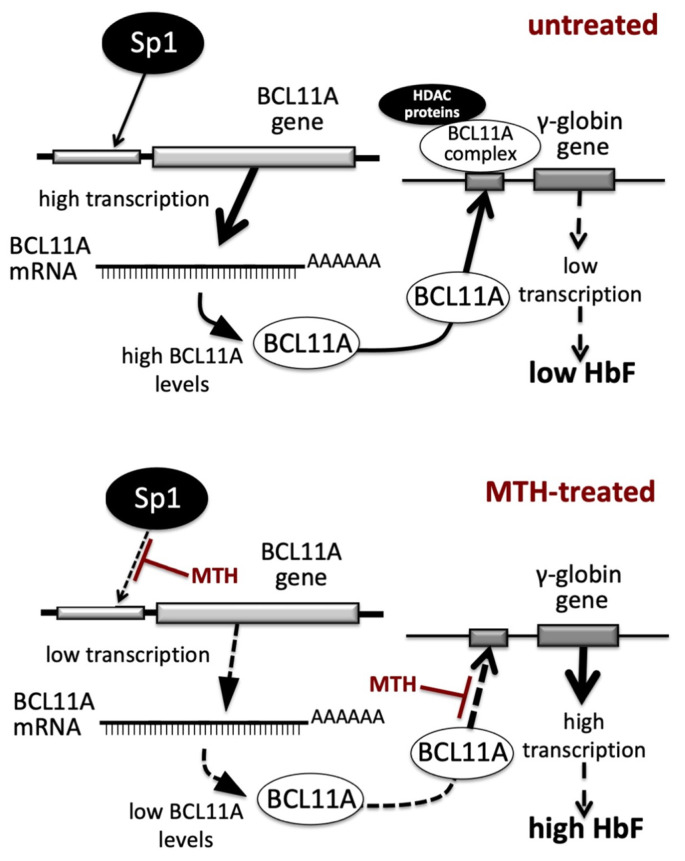
Pictorial representation of the proposed mechanism of action of Mithramycin (MTH) in erythroid cells. Transcription of the BCL11A gene and binding of BCL11A protein to the γ-globin gene promoter in untreated (upper part of the panel) or MTH-treated (lower part of the panel) cells.

## Data Availability

Materials and further information on the data will be freely available upon request to the corresponding author.

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
