# Peer review of "Effects of Mithramycin on BCL11A Gene Expression and on the Interaction of the BCL11A Transcriptional Complex to γ-Globin Gene Promoter Sequences"

_genes, 2023, doi:10.3390/genes14101927_

Round 1

Reviewer 1 Report

Dear Authors,

some things need to be corrected or calryfie.

1) what is the origin of K562 and K562(BCL11A-XL) cella used in this study? there is no information.

2) it is K562 (BCL-11A-XL) cells derived from patients with B-thalassemia? I suppose note, but this is not clear in the manusript. How this cells were cultured? there is only information of K562 cells.

3) Wester blot: what is the total protein per lane used in this techniques? there is no information, also there is no information of protein used as control, references protein.

4) Statistical analysis: what software was used? what test was used to check the normality of distribution? 

5) references: there is the mess in numbering and some typos in this section

6) Figure 2: statement "good correaltion" is absolutely not scientific. It was a Spearman's correlations?? what kind and strongest (p level, r?) In this figures there are data from only one patients without b-thalassemia and one with (panel B and C); this is really small group.

7) in the sections material and methods there is no information of patients, group size; in the text there is infomation under Fig.2 (A) about 12 patients, this group size is very small

In the text minor correction of English is needed

Author Response

Reply to Reviewer #1.

Dear Authors,

some things need to be corrected or clarified.

Answer. Thanks for your work and the suggestions.

Point 1. What is the origin of K562 and K562(BCL11A-XL) cells used in this study? there is no information.

Answer. Thank you for raising this point. The human leukemia K562 cell line was obtained by ATCC, while the K562(BCL11A-XL) clone #12 has been obtained in our laboratory by transfecting K562 cells with the pCDNA3.1-BCL11A-XL vector [Finotti et al., 2015]. In order to clarify this point the following sentence has been included: “The human leukemia K562 cell line [54] has been purchased from the American Type Culture Collection (ATCC, Rockville, Md., USA) ……. despite the fact that expresses at high level BCL-11A, is however inducible by MTH, being therefore useful for studies on possible mechanism(s) of action of this DNA-binding drug [55]” (page 3, lines 96-104). The title of the chapter has been slightly changed in “2.1. Human K562 cell lines and culture conditions”.  

Point 2. It is K562 (BCL-11A-XL) cells derived from patients with B-thalassemia? I suppose not, but this is not clear in the manuscript. How these cells were cultured? There is only information of K562 cells.

Answer. No, the K562 (BCL-11A-XL) cells were not derived from patients with B-thalassemia, but have been obtained in our laboratory by transfecting K562 cells with the pCDNA3.1-BCL11A-XL vector [Finotti et al., 2015] (see also the answer to point #1). The culturing of the K562 cell lines has been described with the following sentence: “Both K562 cell lines have been cultured in humidified atmosphere of …. in vitro cell proliferation was studied by determining the cell number per ml with a Z2 Coulter Counter (Beckman Coulter, Fullerton, CA, USA)” (page 3, lines 104-110).

Point 3. Western blot: what is the total protein per lane used in these techniques? There is no information, also there is no information of protein used as control, references protein.

Answer.  We clarify with the sentence: “An amount of 12 µg of cytoplasmic protein extracts was denatured …” (page 3, lines 133, and legend to Figure 1C, line 215). As far as the internal control, the primary antibody against p70S6K was employed, as stated in line 146 and in the legend to Figure 1C.

Point 4. Statistical analysis: what software was used? what test was used to check the normality of distribution?

Answer. This Section 2.9 has been re-written, indicating the softwares used (page 5, lines 190-195).

Point 5. References: there is the mess in numbering and some typos in this section.

Answer. We deeply apologize for this. We have carefully controlled the numbering of all references, and their correct correspondence with the text. We have corrected typos and style. The newly added references are all red-marked.

Point 6. Figure 2: statement "good correlation" is absolutely not scientific. It was a Spearman's correlations?? what kind and strongest (p level, r?) In this figure there are data from only one patients without beta-thalassemia and one with (panel B and C); this is really small group.

Answer. We thank the reviewer for raising this point. We totally agree with the reviewer’s suggestion. We performed Spearman's correlations and found an inverse strong correlation (rS ​= -0.86667 and p = 0.0025). To clarify this very important point the following sentence has been added: “We found that high relative levels of BCL11A mRNA were associated with low expression of gamma-globin genes; conversely, low relative levels of BCL11A mRNA …  the correlation between the level of BCL11A and the ability of MTH to induce HbF accumulation is very high, being rS = -0.86667 and p (2-tailed) = 0.0025 (Figure 2A).” (pages 5 and 6, lines 221-227).

Point 7. In the sections material and methods there is no information of patients, group size; in the text there is infomation under Fig.2 (A) about 12 patients, this group size is very small.

Answer. This information is important. The recruitment of patients is now reported at page 3, lines 111-122). The section is “2.3. Recruitment of beta-thalassemia patients and in vitro culture of Erythroid Precursor Cells (ErPCs)”. The sentence is:” Patients have been recruited at the Day Hospital Thalassemia and Hemoglobinopathies of Azienda Ospedaliera-Universitaria S. Anna (Ferrara, Italy) …... the patients participating to this study signed an informed consent form in line with the approval of the Ethical Committee in charge of human studies at the University Hospital. The β-thalassemia patients were all transfusion de-pendent and were not under therapy with other HbF inducers” (page 3, lines 117-122). The limit of the small cohort of patients has been discussed at pages 9 and 10, lines 343-349, by including the sentence “Another limit is the number of recruited β-thalassemia patients. In this respect we like to underline that, …. This study should be therefore considered a proof-of-principle regarding possible mechanism(s) of action of a drug of possible interest for beta-thalassemia”.  As far as Figure 2, B and C, the objective of the experiment was to verify the relationship between the BCL11A-XL-mRNA and gamma-globin mRNA at different MTH concentrations. The inverse correlation was confirmed in agreement with Figure 2A.

In conclusion, we believe that, thanks to your work and suggestions, the paper is now much improved in scientific quality and presentation and we hope that it will be considered by you acceptable.

Thanks again for the useful comments and suggestions.

Sincerely,

Alessia Finotti and Roberto Gambari

Reviewer 2 Report

The reviewed manuscript is devoted to an interesting and important topic corresponding to the issue of the journal "Gene". New important results were obtained. Only some technical notes can be made, which could be corrected rather easily. Notes are included to the attached file.

Author Response

Reply to Reviewer #2.

The reviewed article is devoted to an interesting and important topic corresponding to the issue of the journal “Gene”: The possible mechanisms of an antitumor antibiotic mithramycin action on γ-globin gene promoter through transcriptional repressors were proposed. It is very important for β-thalassemia patients, thus this study has an undoubted practical significance for biomedicine.

At the same time, it is possible to make some notes to this paper.

Answer. We thank the reviewer for her(his) positive comments and for the suggestions, all of which are very useful for improving the manuscript.

Point 1. The source of fetal bovine serum is indicated as “FBS; Biowest, Nuaille, F” (line 82). It is better to write “France” instead of “F”.

Answer. Done as suggested (line 107).

Point 2. Figures are usually have to be placed after their first mention in the text, but Figures 5 and 6 are situated a page before their mentions.

Answer. Done as suggested. We verified the Figure location and all of them are placed after their first mention.

Point 3. The inscriptions in Fig. 4 are very small and poorly distinguishable.

Answer. We have enlarged Figure 4. In addition, we have included the sequences of the BCL11A binding site within the Figure legend, in order to help the reader (lines 269-270).

Point 4. The legend to Fig. 6 contains rather long text, which looks like not as a figure description but as a piece of discussion, and part of it better to be moved to the “Discussion” section.

Answer. We agree with this suggestion and we moved almost all of the legend to the text.

The legend to Figure 6 has been modified as follows: “Pictorial representation of the proposed mechanism of action of Mithramycin (MTH) in erythroid cells. Transcription of the BCL11A gene and binding of BCL11A protein to the g-globin gene promoter in untreated (upper part of the panel) or MTH-treated (lower part of the panel) cells”.      

Point 5. Several misprints occur in the manuscript. On line 50, the word “that” is written twice in a row. On line 100, the substance “glycerol” is written as “glicerol”.

Most of these comments are technical and could be corrected rather easily.

Answer. Done as suggested. In addition, we found and corrected other misprints. Thanks again for your work.

In conclusion, we believe that, thanks to your work and suggestions, the paper is now much improved in scientific quality and presentation and we hope that it will be considered by you acceptable.

Thanks again for the useful comments and suggestions.

Sincerely,

Alessia Finotti and Roberto Gambari

Reviewer 3 Report

The manuscript deals with molecular effects of mithramycin, a DNA-binding drug which, along with its anticancer effects, may induce synthesis of HbF, e.g., in erythroid cells from thalassemic patients. The article concerns inhibition of BCL11A gene expression caused by mithramycin, being associated with derepression of g-globin gene in a erythroid cell culture.

The well-designed flow chart included direct qPCR assays of BCL11A gene repression, inhibition of BCL11A synthesis, and mithramycin-mediated inhibition of the BCL11A repressor effect on g-globin gene promoter ( with appropriate . Potentially, this finding may be of some therapeutic significance in thalassemia. 

Materials and Methods: K562 cell line presents a popular in vitro model for molecular hematology studies, and one may be confident to the results shown in the article. However, this cell model of malignant cell population exists for many decades and may be genetically altered in some gene characteristics. A more reliable comparison group should include non-malignant erythroid cells, i.e., those taken from the donor bone marrow, or patients free of thalassemic traits. The results obtained with normal erythroid cell cultures could be more appropriate and convincing.

 Results:

Line 173: When describing relations between BCL11A and MTH-induced HbF expression, one should indicate numerically the correlation quotient r and number of experimental points (Fig.2A).

 In general, the paper presents quite convincing results which may be well interpreted in terms of pharmacogenomics and potentially novel approaches to treatment of Hb-pathies.

Author Response

Reply to Reviewer #3.

Comments and Suggestions for Authors

Point 1. The manuscript deals with molecular effects of mithramycin, a DNA-binding drug which, along with its anticancer effects, may induce synthesis of HbF, e.g., in erythroid cells from thalassemic patients. The article concerns inhibition of BCL11A gene expression caused by mithramycin, being associated with derepression of gamma-globin gene in a erythroid cell culture.

The well-designed flow chart included direct qPCR assays of BCL11A gene repression, inhibition of BCL11A synthesis, and mithramycin-mediated inhibition of the BCL11A repressor effect on gamma-globin gene promoter. Potentially, this finding may be of some therapeutic significance in thalassemia.

Answer. We thank the reviewer for her(his) positive comments.

Point 2. Materials and Methods: K562 cell line presents a popular in vitro model for molecular hematology studies, and one may be confident to the results shown in the article. However, this cell model of malignant cell population exists for many decades and may be genetically altered in some gene characteristics. A more reliable comparison group should include non-malignant erythroid cells, i.e., those taken from the donor bone marrow, or patients free of thalassemic traits. The results obtained with normal erythroid cell cultures could be more appropriate and convincing.

Answer. The reasons for using K562 cells have been presented at page 2. Lines 78-84 (sentence: “The human leukemia K562 cell system was employed as it recapitulates constitutive and induced differentiation and was in the past extensively used for the screening of inducers of embryo-fetal globin genes [6,7,49,50]); we like to underline that the phenotype of K562 cells was routinely checked for expression of erythroid markers such as transferrin receptor and glycophorin A (FACS analyses) as pointed out at page 3, lines 102-104. The ErPCs system was employed as it is a recognized tool for pre-clinical studies of HbF inducers of possible interest for therapeutic protocols for β-thalassemia and other hematological diseases (such as sickle-cell disease) for which increase of HbF production is beneficial [6,7,48,51]”). It should be noted that normal ErPCs do not express at high levels the gamma-globin genes. This was the reason for using ErPCs from beta-thalassemia patients. The limit in using the erythroleukemia K562 cells has been discussed at page 9, lines 337-342).

 Point 3. Results: Line 173: When describing relations between BCL11A and MTH-induced HbF expression, one should indicate numerically the correlation quotient r and number of experimental points (Fig.2A).

Answer. This is an important issue that deserves to be clarified and commented. We thank the reviewer for raising this point. We totally agree with the reviewer’s suggestion. We performed Spearman's correlations and found an inverse strong correlation (rS ​= -0.86667 and p = 0.0025; n = 9). To clarify this very important point the following sentence has been added: “We found that high relative levels of BCL11A mRNA were associated with low expression of gamma-globin genes; conversely, low relative levels of BCL11A mRNA …  the correlation between the level of BCL11A and the ability of MTH to induce HbF accumulation is very high, being rS = -0.86667 and p (2-tailed) = 0.0025 (Figure 2A).” (pages 5 and 6, lines 221-227).

Point 4. In general, the paper presents quite convincing results which may be well interpreted in terms of pharmacogenomics and potentially novel approaches to treatment of Hb-pathies.

Answer. We thank the reviewer for her(his) positive comments and we are grateful for the suggestions.

In conclusion, we believe that, thanks to the work provided by the Editors and the Referees the paper is now much improved in scientific quality and presentation and we hope that it will be considered acceptable for publication on the MDPI Journal GENES.

Thanks again Editors and Reviewers for the useful comments and suggestions.

Sincerely,

Alessia Finotti and Roberto Gambari

Round 2

Reviewer 1 Report

Dear Authors,

I recommended this paper to publish, all corrected, all satisfied.